# Clinical and Pathological Profile of Children and Adolescents with Osteosarcoma

**DOI:** 10.3390/diagnostics15030266

**Published:** 2025-01-23

**Authors:** Andrei Ivan, Elena Cojocaru, Paul Dan Sirbu, Dina Roșca Al Namat, Ștefan Dragoș Tîrnovanu, Lăcrămioara Ionela Butnariu, Jana Bernic, Valentin Bernic, Elena Țarcă

**Affiliations:** 1Department of Surgery II—Pediatric Surgery, “Grigore T. Popa” University of Medicine and Pharmacy, 700115 Iasi, Romania; ivan.andrei@umfiasi.ro (A.I.); dina.rosca-al.namat@umfiasi.ro (D.R.A.N.); tarca.elena@umfiasi.ro (E.Ț.); 2Department of Morphofunctional Sciences I—Pathology, “Grigore T. Popa” University of Medicine and Pharmacy, 700115 Iasi, Romania; 3Department of Orthopedics and Traumatology, Surgical Science (II), Faculty of Medicine, “Grigore T. Popa” University of Medicine and Pharmacy, 700115 Iasi, Romania; paul.sirbu@umfiasi.ro (P.D.S.); stefan-dragos.tirnovanu@d.umfiasi.ro (Ș.D.T.); 4Department of Medical Genetics, Faculty of Medicine, “Grigore T. Popa” University of Medicine and Pharmacy, 700115 Iasi, Romania; ionela.butnariu@umfiasi.ro; 5Discipline of Pediatric Surgery, “Nicolae Testemițanu” State University of Medicine and Pharmacy, 2004 Chisinau, Moldova; jana.bernic@usmf.md; 6Department of Surgery II, “Saint Spiridon” Hospital, University Street No. 16, 700115 Iasi, Romania; bernic.valentin@email.umfiasi.ro

**Keywords:** adolescent, clinical profile, malignant bone tumour, osteosarcoma, prognostic factors

## Abstract

**Introduction:** Osteosarcoma (OS) is the most common type of primary malignant bone and cartilage tumour. Because of the remarkable developments in technology, remarkable progress has been made in the medical field regarding the diagnosis and management of OS patients. The aim of the study is to describe the clinical and pathological profile of paediatric patients with osteosarcoma and to identify potential prognostic factors for an unfavourable outcome in our country. **Methods:** We conducted a retrospective study of all children and adolescents with musculoskeletal tumours diagnosed and treated at our tertiary Orthopaedic Department for a period of 10 years. **Results:** A group of 65 children and adolescents with osteosarcoma who benefited from diagnosis, neoadjuvant, adjuvant and surgical treatment in the Emergency Clinical Hospital for Children “Sfânta Maria” Iasi, România, was analysed. The average age at the time of diagnosis was 12.9 years. The analysis revealed a higher frequency for male patients in the case of femur and tibia locations and a significantly higher frequency of osteosarcoma in the scapula and clavicle in female patients, while OS in the humerus was found only in male patients (χ^2^ = 19.46, *p* = 0.0149). The most frequent histopathological subtype was osteoblastic osteosarcoma, but there was no significant correlation with the gender or the age of the patients (χ^2^ = 0.73, *p* = 0.863 and χ^2^ = 0.843, *p* = 0.839). The results indicated instead a significantly (*p* = 0.0185) lower age values of patients with undifferentiated osteosarcomas, the average age being 9.4 years ± 2.1 SD. After performing a multivariate logistic regression analysis for the risk of death based on clinical parameters, we found that high tumoural grading increases the risk of death 2.8 times, pleomorphic histological subtype increases the risk of death 3.5 times, and stage IV TNM increases this risk 5.9 times. **Conclusions:** For the north-eastern geographical part of Romania, the epidemiological and clinical profile of a child with osteosarcoma is a 13-year-old boy with a femoral or tibia tumour or a 12-year-old girl with a femoral, tibia, scapula or clavicle tumour, both coming from a rural area. The tumour has around 12 cm diameter and is a differentiated osteoblastic osteosarcoma. The survival rate at 10 years is 63%. Tumour grading, histological subtype and TNM staging significantly influence the probability of death and could be important prognostic parameters for patients with osteosarcoma.

## 1. Introduction

Tumours of the musculoskeletal system present a variety of challenges, from diagnosis to treatment and long-term sequelae. Osteosarcoma (OS) is the most common type of primary malignant bone and cartilage tumour and is defined by the presence of malignant mesenchymal cells producing osteoid or immature bone [1].

Osteosarcoma in children and adolescents is rare and accounts for approximately 3–5% of all malignant tumours diagnosed in children up to 15 years of age; in a higher percentage, approximately 7–8% is found among adolescents aged between 15 and 19 years [2]. The overall annual incidence rate is 2–4:1,000,000 cases per year; the incidence of osteosarcoma in children and adolescents increased from the 1970s to the 2000s but decreased in adults [2,3,4]. Osteosarcoma does not appear to be linked to an obvious cause, and its aetiology is not well understood. It appears at increased rates in young people with Paget’s disease of bone, taller individuals, after therapeutic radiation, or in some cancer predisposition syndromes [4]. Environmental components are not strongly associated with osteosarcoma risk. This pathology affects both sexes, with a slight male preponderance, and the age at presentation ranges from 10 to 25 years of age [5].

Thanks to the remarkable development of technology, remarkable progress has been made in the medical field regarding the diagnosis of patients with osteosarcoma by improving radiological (computed tomography, magnetic resonance) and anatomopathological (immunohistochemistry, molecular biology) diagnostic methods. The delay in diagnosing this condition is mainly due to the lack of health education among the population and the awareness of the possibility of such a diagnosis among adolescents. Although significant progress has been made over the past century, most patients have a poor prognosis due to the spread of the disease to other organs. Therefore, it is essential to find prognostic factors with high power to predict the outcome for paediatric patients with osteosarcoma. The current survival rate is considerably higher compared to the year 2000 due to the development of oncological orthopaedics, medical–surgical equipment and adjuvant therapy [6]. Little information is available regarding osteosarcoma in children because these cases are typically grouped with adolescent and young adult cases due to the rarity of the disease [7].

The aim of the study is to describe the most frequent clinical profile of paediatric patients with osteosarcoma based on a 10-year case history in our country, considering the diagnostic progress of recent years. The present research also attempts to identify potential prognostic factors in paediatric patients diagnosed and treated with osteosarcoma.

## 2. Materials and Methods

We conducted a retrospective study of all children and adolescents with musculoskeletal tumours diagnosed and treated at our Orthopaedic Department. Prior to data extraction, approval was granted by the Ethics Committee of “Saint Mary” Emergency Children’s Hospital, and informed consent was obtained from the authorised persons/parents. Inclusion criteria for this analysis consisted of patients who were diagnosed and treated for osteosarcoma (confirmed by biopsy and pathological study) in our tertiary care centre, with no previous oncological treatment for another type of oncological condition, from January 2013 to December 2022. Seventeen patients with other than OS musculoskeletal tumours (bone metastases, benign bone tumours and chronic osteomyelitis (initially suspected as an osteolytic bone tumour)), three patients with incomplete medical records who were initially operated elsewhere and two patients with multiple metastases for whom palliative treatment was established were excluded from the analysis (Figure 1). Patient data were extracted from the computerised database of the hospital and were statistically processed. All patients included in the study were investigated by clinical exam, demographic data, medical history, the location of the primary tumour, the presence of metastases at diagnosis, the size of the tumour, complete blood count, serum biochemistry, plain X-ray of the chest and the site of the primary tumour, computed tomography (CT) of the chest, radioisotope bone scan and a CT scan or magnetic resonance imaging (MRI). These assessments were performed preoperatively and at regular intervals thereafter, during, and after completion of therapy. Normal neutrophil values in children are between 13,000 and 38,000/μL. Mild neutropenia is considered when the absolute value of neutrophils varies between 1000 and 1500/μL. Moderate neutropenia is considered when the absolute value of neutrophils varies between 500 and 1000/μL, and severe neutropenia is when the absolute value of neutrophils is lower than 500/μL [8].

The American Joint Committee on Cancer (AJCC) and the Union for International Cancer Control (UICC) described the TNM staging system in 2010, which we also used in our clinic during the analysed period [9]. All bone tumours included in this study were reclassified in accordance with the WHO Classification of Tumors: Soft Tissue and Bone Tumors, 2020 [10].

The study group was divided into two groups according to the age of the patients, the age of 12 years being used as a reference value, similar to other studies [7,11,12]. The research employed the age of 12 as a reference value for the categorisation of paediatric patients with osteosarcoma since it is a significant variable for stratifying the probability of osteosarcoma incidence. Children under the age of twelve and adolescents between the ages of twelve and eighteen are classified according to these criteria for paediatric osteosarcoma patients.

### Statistics

Statistical analyses of the variables of interest were performed using SPSS v.29 software. Continuous variables were reported as mean with standard deviation. Comparisons between analysed groups were performed using Student’s *t*-test, ANOVA, Kruskal–Wallis or Mann–Whitney U Test for continuous variables. The homogeneity of the series was checked regarding the statistical differences between the variances of the series by the Levene test. Correlations between certain parameters were tested using the Pearson test by evaluating the correlation coefficient r. Qualitative variables were presented as absolute (*n*) and relative (%) frequencies, and comparisons between groups were made based on the results of the non-parametric M–L, Yates or Pearson chi-square tests. Following univariate analyses, a mathematical model was identified for evaluating the chance of occurrence (probability of occurrence) of complications or deaths. The identified mathematical model was verified by Omnibus Tests of Model Coefficients, Nagelkerke R Square, and Hosmer–Lemeshow Test. The level of significance calculated within the tests used (*p*-value) was considered significant for values *p* < 0.05.

## 3. Results

### 3.1. Demographic and Clinical–Pathological Characteristics of Osteosarcomas in Children and Adolescents

A group of 65 children and adolescents with osteosarcoma who benefited from diagnosis, neoadjuvant, adjuvant and surgical treatment in the Emergency Clinical Hospital for Children “Sfânta Maria” in Iasi, România, was analysed. The average age at the time of diagnosis was 12.9 years (3.63 SD). The age of the cases presented a normal distribution with a median value of 13 years (between 5 and 18 years), which highlights a frequency of 50% of osteosarcoma cases in patients older than 13 years. The male/female ratio in the analysed group was 1.7. Demographic and pathological characteristics of the studied group are presented in Table 1, Table 2 and Table 3. Since the analysed variable age (years) does not follow a normal distribution (the Shapiro–Wilk test returned a statistically significant result), our data in Table 1 are presented using the median values and quartiles. Table 2 describes the diameter of the primary tumour (cm), measured on CT scans, MRI or X-ray imaging, depending on the age, gender and location of the tumour.

For the studied group, the main symptoms at presentation were attributed by the patient and relatives to trauma. The main symptoms were pain (93.8%), functional impotence (87.7%) and swelling (100%). Figure 2, Figure 3 and Figure 4 present the clinical, imaging and intra-operative appearance of the humeral and femoral OS for two of our patients.

In the study group, the most frequent localisation of OS was at the level of the long bones. The diameter of the primary tumour was not significantly different depending on the age or gender of the patient. In terms of tumour size, 69.2% of the cases presented a tumour diameter greater than 10 cm, and 44.6% was even greater than 15 cm; the largest tumour sizes were recorded at the level of the tibia (14.1 ± 3.7 cm).

The results also indicated a significantly higher frequency of OS in the scapula and clavicle in female patients, while OS in the humerus was found only in male patients (χ^2^ = 19.46, *p* = 0.0149). The analysis revealed a higher frequency for male patients in the case of femur and tibia locations.

The analysis of the frequency of cases according to the location of the primary tumour indicated that osteosarcoma at the level of the tibia is more frequent at the age of less than 12 years (21.05% vs 33.3%), while at the level of the femur, the frequency is slightly higher in the age group over 12 years (55.2% vs 51.85) (χ^2^ = 4.183, *p* = 0.65182, r = 0.0091). The frequency of cases showing the localisation of osteosarcoma at the level of the humerus was 10.53% in adolescents (over 12 years old) and 3.7% in children under 12 years old (Table 3).

In the analysed study group, the most frequent histopathological subtype of osteosarcoma was osteoblastic osteosarcoma (Table 3). The analysis regarding the association of the histological subtype of osteosarcoma with the gender of paediatric patients did not indicate a significant correlation (χ^2^ = 0.73, *p* = 0.863).

Instead, there was a significant association between the age of paediatric patients and the degree of tumour differentiation (Table 3). Considering the significant association between the age of the patients and the degree of differentiation that emerged from the non-parametric analysis, we concretely quantified the age differences. This was possible considering that the pretest analysis indicated that there were no significant differences between the age variances according to the degree of differentiation. The results indicated significantly (*p* = 0.0185) lower age values in the case of undifferentiated osteosarcomas, the average age being 9.4 years ± 2.1 SD.

The TNM staging of paediatric osteosarcoma in the study group is presented in Table 3. In the analysed group, there was no significant association between the gender of paediatric patients and the TNM stage (χ^2^ = 2.11, *p* = 0.549).

### 3.2. Biological Markers

Biological markers (detected at the time of diagnosis) of OS patients depending on the age group are presented in Table 4.

Alkaline phosphatase (ALP) is usually elevated in patients with osteogenic sarcomas. In the analysed group, 75% of the paediatric patients presented ALP values higher than 703 U/L, and the average value was 991.7 U/L. The ALP values did not show significant differences between the children and adolescents included in the study group, nor according to the gender of the patients (Table 4).

In paediatric patients with osteosarcoma, lactic dehydrogenase (LDH) and erythrocyte sedimentation rate (ESR) values were significantly increased but did not present significant differences between children and adolescents included in the study group (Table 4). Serum LDH was significantly higher in patients with metastatic disease at presentation than in patients with localised disease.

In the analysed group, neutrophil values were lower than 1500/μL in all paediatric patients with osteosarcoma (normal values in children are between 13,000 and 38,000/μL). Moderate neutropenia is considered when the absolute value of neutrophils varies between 500 and 1000/μL, and severe neutropenia is when the absolute value of neutrophils is lower than 500/μL [8]. The neutrophil values showed significant differences between the children and adolescents included in the study group (Table 4). Thus, the values of neutrophils in adolescents were significantly higher compared to those of children younger than 12 years old. The results obtained following the evaluation of neutrophil values in paediatric patients with osteosarcoma revealed a significant positive correlation (r = 0.321, *p* = 0.037) with their age.

Normal lymphocyte values for children are between 3000 and 9500 μL. Lymphocyte values were low in all patients and did not present significant differences between children and adolescents included in the study group, nor according to their gender (Table 4). Platelet values were low in all patients with osteosarcoma, being on average 68,500/IU, without significant differences between the two age groups or between the two genders.

### 3.3. Survival

The survival over the 10-year period in which the patients were analysed was 63.1%. The survival rate evaluated in the first 94 months shows a significant difference depending on the age group (Log-Rank Test: LRT = 3.26, *p* = 0.0039). The survival does not show significant differences according to the gender of the patients.

### 3.4. Prognostic Factors Based on Clinical Parameters

Next, we performed a multivariate logistic regression analysis to analyse the risk of death based on the clinical parameters age, gender, location, tumour grading, histological subtype and TNM staging. The “ENTER” model was applied in which all independent variables (clinical parameters) were included in one step. Based on the constant model (without predictors), 75.7% of cases can be correctly predicted from the point of view of the occurrence of complications. Also, in this step, the contribution of each independent variable to the improvement of the model is evaluated. The probability of observing the real situation is then evaluated by applying the ratio probability test. This tests the difference −2 LL (likelihood ratio) between the full model with predictors and the original model without predictors (null model) based on the chi-square test (Omnibus Tests of Model Coefficients). The results indicate that the model can correctly evaluate a significant number (χ^2^ = 19.15, *p* = 0.0085) of cases (−2 LL = 66.45). The model calibration results for the analysed data (Hosmer–Lemeshow test) (*p* = 0.118) indicate that the estimated frequency of complications is not significantly different from that estimated by the model, thus resulting in the fact that the generated model is appropriate. The proportion of correctly classified cases following the introduction of predictors was 73.8%. This result demonstrates an improvement in the model; basically, the model with predictors is significantly better. The Wald statistic and associated probabilities present a significance level for each predictor. The unstandardised coefficients B (first column) keep the units of measure of the variables and express the variation of the dependent variable according to the independent variables. To compare the coefficients and establish a hierarchy of the predictive power of each predictor, it is necessary to consider Exp(B), which estimates the odds ratio regarding the occurrence of death. The contribution of each variable to the occurrence of the event will be evaluated. The “B” values are the logistic coefficients that can be used to create a predictive equation (Table 5).

Among the clinical parameters, only tumour grading, histological subtype and TNM staging significantly influence the probability of death.

## 4. Discussion

Osteosarcoma is a malignant intraosseous tumour with a high degree of malignancy formed by tumour cells that produce osteoid, bone or chondroid material [13]. Osteosarcoma is one of the most common primary malignant bone tumours of the long bones with an increased frequency in children and adolescents [14,15]. It represents approximately 19% of all malignant bone tumours [2]. Although some studies and registries [16,17] report the pelvis as the preferred location for OS, followed by the diaphysis of the long bones, our study reveals a preferred location in the femur and tibia (80%), followed by the humerus, and only 1.54% of cases were located in the pelvis. The most frequent location, according to other studies, is at the level of the distal metaphysis of the femur (where the growth cartilage is responsible for 70% of the development of the femur) with a percentage of 30%, followed by the proximal tibial metaphysis (responsible for 55% of the growth tibia) with a percentage of 15% and the proximal humeral metaphysis (where the growth cartilage is responsible for 80% of the development of the humerus) with a similar percentage, 15% [18,19]. Patients with tumour localisation in the femur or tibia had a better survival rate in our study. Although small differences can be observed between the mean ages of the patients included in the study, we definitively validated the fact that the location of the osteosarcoma was not associated with the age of the children, despite the fact that there are studies that show a significant association between the location of the primary tumour and the age of paediatric patients with osteosarcoma [20,21].

The pathology is closely related to several factors: age, sex, race, height, genetic factors, congenital bone abnormalities and some cancer predisposition syndromes like Li-Fraumeni syndrome, hereditary retinoblastoma, Rothmund-Thomson type II, Werner and Bloom syndrome or Diamond-Blackfan anaemia [22,23,24]. The main predisposing factors in the appearance of the tumour are repeated traumas in the past, benign bone tumours with the potential for malignancy (e.g., tumour with myeloplaxes) and multiple irradiations performed for other conditions. We found a median age at diagnosis of 12.9 years, with a 1.7 male predominance, similar to other studies reporting a male-to-female prevalence ratio of 1.4:1 [25,26]. Although some studies have reported that females less than 15 years of age are more affected than males in the same age group, we did not find this association [2,26]. Female patients have higher survival rates than males in many studies [21,26,27,28], but the survival rate did not show significant differences according to the gender of the patients or the location of the tumour in our study. Mirabello et al. demonstrated a higher survival rate when osteosarcoma occurs in the short bones and the poorest survival rate for osteosarcoma of the pelvic region and vertebral column for all ages [2].

Rapid bone growth and hormonal changes at puberty may be involved in osteosarcoma aetiology, and we demonstrated a lower average age of OS diagnosis in girls (12.4 years) compared to boys (13.2 years), in accordance with the onset of puberty faster in girls than in boys. Although osteosarcoma can occur in any bone in the body, the most common are the sites of rapid bone growth, the metaphysis of long bones, which reinforces the relationship between rapid bone growth and osteosarcoma formation, because of increased vulnerability at the metaphysis due to the high cell turnover during puberty [4,27].

When osteosarcoma occurs in older people, there is a possibility that a pre-existing benign tumour may become malignant (e.g., Paget’s disease) and there may be bone dysplasia or bone infarction. The age-adjusted incidence rate was 1.9 per million for those aged 0 to 9 years, 6.7 per million for those aged 10 to 24 years, 1.9 per million for those aged 25–59 years and 2 per million for those aged ≥ 60 years [28,29].

The contribution of toxins and environmental exposures to osteosarcoma or other cancers of children is not fully understood; we found a higher frequency of OS in children from rural areas (69.2%), and this association needs further investigation. Regarding demographic data, there are no precise statistics for the exact number of children living in villages in Romania at any given time, but official data from censuses and official reports indicate a significant trend of rural-to-urban migration [30]. According to data published by the National Institute of Statistics, approximately 45% of Romania’s population lived in rural areas in 2021, and a large part of this population includes children.

The COVID-19 pandemic had negative implications on many levels, both in the socio-economic and medical fields, with the decrease in the rate of early diagnosis of certain diseases, including cancer. The presentation of children to the doctor was affected due to the difficulties faced by the parents on a social, educational or financial level [31,32]. Understanding the contribution of certain environmental factors to the occurrence of osteosarcoma is very difficult due to the rarity of this malignancy and the fact that exposure to a single environmental factor is unlikely to be the primary cause [33].

Contrary to benign tumours, which appear insidiously and are asymptomatic, malignant bone tumours are frequently associated with pain, functional impotence and swelling. Unfortunately, parents and children often associate this symptomatology with an injury during sports class or playing, and thus, the diagnosis is made late. Although pain is usually the first symptom and swelling appears later [33], all our patients had obvious swelling at admission, which indicates a late diagnosis. The majority of our patients declared that pain was initially of low or medium intensity but quickly became severe. Pain has a maximum intensity after exertion and at night, without being calmed by rest. It has an osteocope character, with the child always having the impression that the bone is breaking. Functional impotence is relative and is more marked if the lower limbs are affected, with lameness occurring early in the evolution of the condition. The swelling initially appears as a tumour mass in close contact with the bone, which is preceded by pain. An increase in the respective region is noted due to a rapid involvement of the soft parts and the development of collateral circulation. In advanced forms, the skin can ulcerate, and the externalised tumour formation becomes infected. Patients may also present with weight loss, fever, fatigue, or malaise [34]. Dissemination is strictly hematogenous, so locoregional adenopathy does not occur. In terms of tumour size, 69.2% of the cases presented a tumour diameter greater than 10 cm, and 44.6% was even greater than 15 cm; the largest tumour sizes were recorded at the level of the tibia (14.6 ± 3.3 cm). A possible explanation for the large tumour size at presentation may be the late addressability of the cases, especially since most of our patients came from the countryside.

Inflammation caused by tumours has been recognised as a characteristic of oncological conditions. The interaction between the systemic inflammation and local immune response plays a vital role in OS progression and the evolution of the patient. Thus, certain biological tests that show inflammation can be used to predict tumour prognosis. Measuring the number of lymphocytes, neutrophils, and platelets will evaluate the systemic inflammatory responses. Ouyang et al. demonstrated that low neutrophile and platelet numbers are significantly associated with the prognosis of OS patients [35]. In our study, the values of the inflammatory parameters (neutrophils, lymphocytes, platelets) were low in all patients, and a significant positive correlation of low neutrophil number with age under 12 years was demonstrated (r = 0.321, *p* = 0.037). In our analysed group, neutrophil values were lower than 1500/μL in all paediatric patients with osteosarcoma, and the difference between children and adolescents was significant (795.9 versus 929.3/µL, *p* = 0.035).

The development of cancer may be impacted by the existence of tumour-infiltrating lymphocytes. Tumour-suppressive or tumour-promoting lymphocytes are the two basic categories into which tumour-infiltrating lymphocytes fall. Interestingly, the quantity of these cells may be a predictor of overall survival and the effectiveness of treatment [36]. The immune microenvironment’s makeup inside a tumour is a major factor in influencing the prognosis of patients as well as how well they respond to immunotherapy treatments. However, immune cell infiltration’s prognostic usefulness differed depending on the type of malignancy [37].

There are few studies that analyse the prognostic value of serum inflammatory biomarkers in osteosarcoma patients [38,39,40,41]. Jettoo et al. showed that a high level of ESR is associated with poor prognosis in patients with osteosarcoma [36], but another study did not demonstrate a relationship between high values of ESR and survival in children with osteosarcoma of the extremities, although patients with pathological ESR had a higher risk of local recurrence [38]. A recent meta-analysis shows that elevated serum LDH is associated with a lower event-free survival rate, so serum LDH could be a prognostic biomarker for OS patients [42]. Our study demonstrated a high level of ESR and LDH in all patients with OS. Also, serum LDH was significantly higher in patients with metastatic disease at presentation than in patients with localised disease.

Alkaline phosphatase is elevated in 60% of patients with osteogenic sarcomas in some reports [38,43]. A return to normal of alkaline phosphatase indicates the effectiveness of cytostatic or surgical treatment. When alkaline phosphatase is elevated, it reflects the size of the tumour mass. The persistence of elevated values after the application of the treatment indicates its ineffectiveness. ALP values are especially important in following the evolution of patients; they decrease and reach normal levels after resection of the tumour formation and increase in cases of recurrence or metastases [31]. In general, for children up to 12 years of age, the normal value of the alkaline phosphatase analysis is below 500 U/L. For female children aged 12 to 15 years, the considered normal value for ALP is below 420 U/L, while for male children of the same age, the upper limit for normal ALP values is 750 U/L. In male adolescents, the reference value considered normal for the ALP test is below 260 U/L. In our analysed group, it is noted that 75% of the paediatric patients presented ALP values higher than 703 U/L, and the average value was 991.7 U/L. This aspect confirms once again that ALP values are an important marker in paediatric osteosarcoma in conditions where normal ALP values show differences both according to the gender of the people and according to their age. However, with the pathological values found in osteosarcoma, these inclusion criteria are no longer valid; the ALP values did not show significant differences between the children and adolescents included in the study group (*p* = 0.418), nor according to the gender of the patients.

Survival rates vary by age, gender, location of the tumour and disease stage. For children, these rates range from 55 to 75% in general, but lower rates (19–39%) have been observed in Slovakia, Estonia, and Denmark [19,44]. In the US, the relative 5-year survival rate for young-onset osteosarcoma was 61.6% [2]. Considering the fact that age is an important variable for stratifying the risk of osteosarcoma occurrence, the analysis was carried out using the age of 12 years as a reference value for the grouping of paediatric patients with osteosarcoma. This classification of paediatric patients with osteosarcoma corresponds to the classification of the child (age less than 12 years) and the adolescent (age between 12 and 18 years). The survival rate in our study, evaluated in the first 94 months, showed a significant difference depending on the age group (*p* = 0.0039), with better survival in the group under 12 years of age, similar to the findings of Cole et al. [7,8,9]. These data suggest that the youngest pre-pubertal cases of osteosarcoma may have a different aetiology compared to young adult cases, and that is why they must be evaluated separately.

Since the introduction of chemotherapeutic agents in the mid-1970s and the subsequent advent of neoadjuvant chemotherapy, the 5-year survival rate has improved considerably, reaching 66–82% [7,27,45,46]. At baseline, children and adolescents with localised osteosarcoma have an overall survival rate of 70% at five years and <30% for patients with metastases [44]. The overall survival rate at 10 years in our study was 63%. After performing a multivariate logistic regression analysis to analyse the risk of death based on clinical parameters, we found that high tumoural grading increases the risk of death 2.8 times, pleomorphic histological subtype increases the risk of death 3.5 times, and stage IV TNM increases this risk 5.9 times. Although there are several studies with similar results [35,47,48,49], the present study estimates the risk of death with the help of a regression model. This study suggested that tumour grading, histological subtype and TNM staging significantly influence the probability of death and could be important prognostic parameters for patients with osteosarcoma.

Advances in the treatment of primary osteogenic sarcoma of the extremities and the demonstrated major impact of multi-agent adjuvant chemotherapy in randomised trials have significantly improved relapse-free survival [50,51]. Advances in understanding the genetics and the biology of the disease have provided the foundation for a new wave of innovative clinical trials of targeted therapies, which use treatment directed at the intrinsic molecular biology of osteosarcomas or at antigens ubiquitously expressed on the surface of diverse tumours [52,53,54,55,56]. Highlighting the combination of the expression of some biomarkers with the response to mono-therapy or combined therapy in the case of osteosarcomas in children can explain a series of clinical–evolutionary aspects and lead to better clinical–therapeutic management of these cases [54,56].

Limitation of the study: Although this is the largest cohort of patients with OS analysed in a single study in our country, the generalisation of the results to the whole population of Romania or other countries must be carried out with caution. Another limitation is that the study was retrospectively performed throughout one decade on a small number of patients. Oncology is an evolving field, and there have been significant changes in practice in recent years; medical knowledge and access to technology are evolving, and this can lead to the appearance of biases in a retrospective study over a long period of time.

Future directions: Future pathological, genetic, clinical and laboratory studies and meta-analyses should be carefully conducted to demonstrate that the complex hormonal changes that occur during puberty, as well as tumour grading, histological subtype and TNM staging, have a role in the appearance and prognosis of osteosarcoma.

## 5. Conclusions

For the north-eastern geographical part of Romania, the most frequent epidemiological and clinical profile of a child with osteosarcoma is a 13-year-old boy with a femoral or tibia tumour or a 12-year-old girl with a femoral, tibia, scapula or clavicle tumour, both coming from a rural area. The tumour has around 12 cm diameter and is a differentiated osteoblastic osteosarcoma. In order to improve the early diagnosis and the implicit survival rate, it is imperative to know this condition and to be aware of the risk of its occurrence in children and adolescents. It is also essential to find prognostic factors with high power to predict the outcome for paediatric patients with osteosarcoma. The interplay between the local immune response and systemic inflammation plays a vital role in cancer progression and patient survival. Thus, inflammatory parameters are strong candidates for predicting tumour prognosis. Evaluation of neutrophils, lymphocytes, and platelets can help understand systemic inflammatory responses. Tumour grading, histological subtype and TNM staging significantly influence the probability of death and could be important prognostic parameters for patients with osteosarcoma.

## Figures and Tables

**Figure 1 diagnostics-15-00266-f001:**
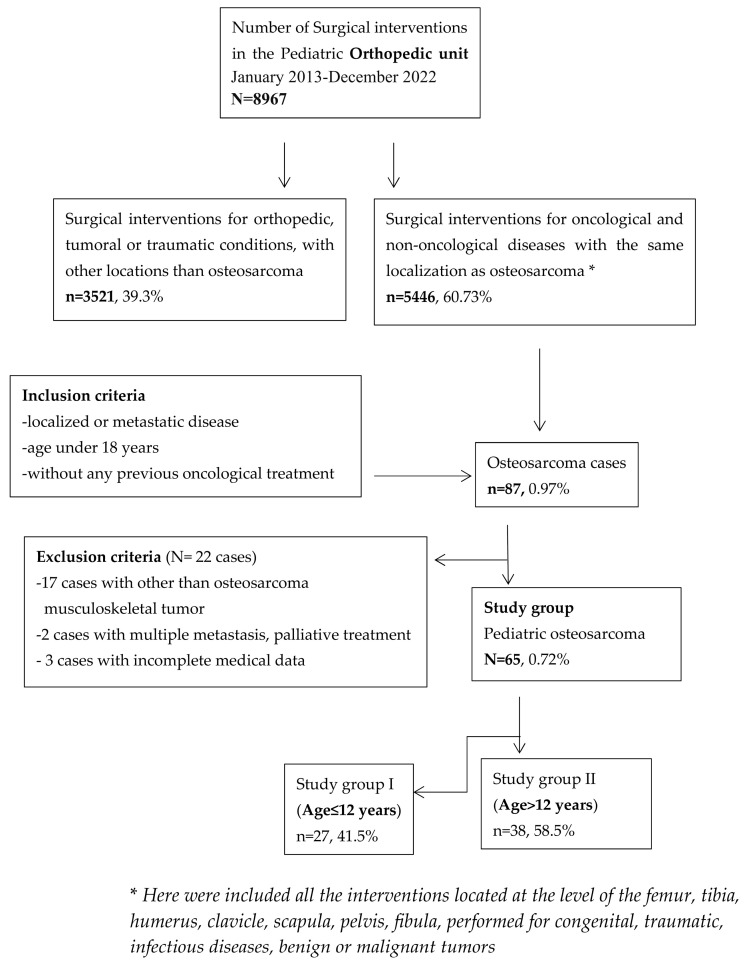
Flowchart of selected patients for the study.

**Figure 2 diagnostics-15-00266-f002:**
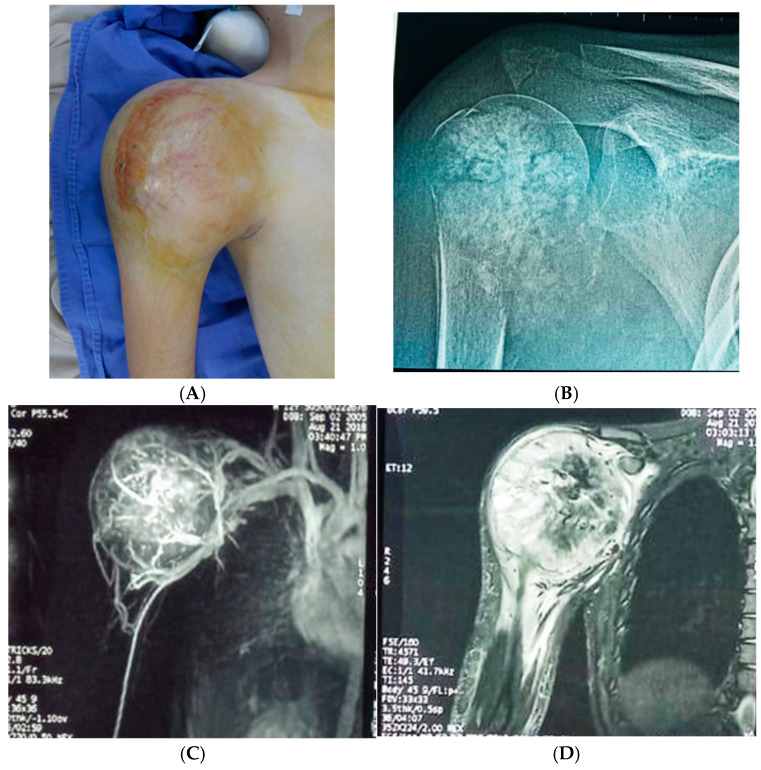
Humeral osteosarcoma, 16-year-old male patient, 14 cm diameter tumour; (**A**) clinical aspect showing proximal humeral tumour, with a central scar—the site of the biopsy; (**B**) humeral X-ray showing periosteal destruction, infiltration in adjacent tissues and bone apposition; (**C**) CT scan demonstrating the rich vascularisation of the tumour; (**D**) MRI image with central tumour necrosis.

**Figure 3 diagnostics-15-00266-f003:**
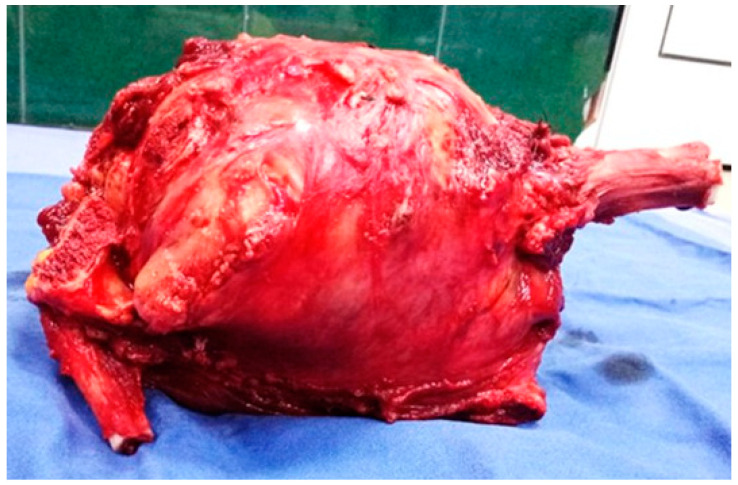
Resection piece containing humeral tumour, proximal half of the humerus, part of the scapula and clavicle—Patient in Figure 2.

**Figure 4 diagnostics-15-00266-f004:**
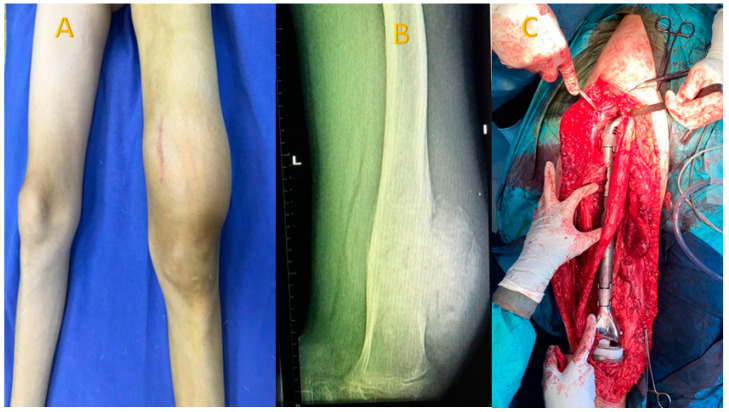
Femoral OS in a 14-year-old male patient, 13 cm diameter tumour; (**A**) clinical aspect showing distal femoral tumour, with a central scar—the site of the biopsy; (**B**) femoral X-ray showing periosteal destruction and infiltration in adjacent tissues; (**C**) intraoperative aspect after removal of the tumour, the femur, and implantation of the articulated femur prosthesis.

**Table 1 diagnostics-15-00266-t001:** Descriptive statistics of the analysed patients (the age distribution of patients according to gender and social environment).

Variable		No. of Cases	Min	Max	Median Age (Years)	Q25	Q75
**Total**		65	4.0	18.0	13.0	10.0	16.0
Gender (*p* = 0.436)	Male (63.1%)	41	4.0	18.0	13.0	11.0	16.0
	Female (36.9%)	24	5.0	18.0	12.5	9.5	16.0
Social environment (*p* = 0.395)	Rural (69.2%)	45	5.0	18.0	13.0	11.0	16.0
	Urban (30.8%)	20	5.0	18.0	12.0	10.0	16.0

**Table 2 diagnostics-15-00266-t002:** Diameter of the primary tumour (cm), measured on CT scans, MRI or X-ray imaging.

	No. of Cases	Mean Diameter (cm)	Std. Dev.	*p*-Value
Total	65	12.3	4.3	-
**Gender**				
Male (M)	41	12.6	4.3	0.473
Female (F)	24	11.8	4.4	
**Age group (child/teenager)**				
<12 years	27	12.7	4.5	0.529
≥12 years	38	12.1	4.2	
**Location of the primary tumour**				
Femur	35 (12 F + 23 M)	13.6	3.3	<0.01
Tibia	17 (8 F + 9 M)	14.1	3.7	
Humerus	5 (0 F + 5 M)	6.8	1.5	
Clavicle	3 (2 F + 1 M)	4.7	0.6	
Scapula	3 (2 F + 1 M)	7.0	2.0	
Pelvis	1 (0 F + 1 M)	5.0	1.8	
Fibula	1 (0 F + 1 M)	12.0	2.6	

**Table 3 diagnostics-15-00266-t003:** Pathological characteristics of the tumour depending on the age of the patients.

		Total	<12 Years	≥12 Years	Pearson Chi-Square
No. of Patients		65	27	38	
Location of the primary tumour	Femur	35	14 (51.85%)	21 (55.26%)	χ^2^ = 4.183 *p* = 0.652
	Tibia	17	9 (33.33%)	8 (21.05%)	
	Humerus	5	1 (3.70%)	4 (10.53%)	
	Clavicle	3	1 (3.70%)	2 (5.26%)	
	Scapula	3	2 (7.41%)	1 (2.63%)	
	Pelvis	1	0 (0.00%)	1 (2.63%)	
	Fibula	1	0 (0.00%)	1 (2.63%)	
Histopathological subtype of osteosarcoma	Osteoblastic	44	19 (70.37%)	25 (65.79%)	χ^2^ = 0.843 *p* = 0.839
	Chondroblastic	17	7 (25.93%)	10 (26.32%)	
	Pleomorph	3	1 (3.70%)	2 (5.26%)	
	With clear cells	1	0 (0.00%)	1 (2.63%)	
Tumour grading	High grade	47	17 (70.83%)	30 (73.17%)	χ^2^ = 0.731 *p* = 0.693
	Intermediate grade	17	7 (29.17%)	10 (24.39%)	
	Periostal–low grade	1	0 (0.00%)	1 (2.44%)	
Degree of osteosarcoma differentiation	Well	8	2 (7.41%)	6 (15.79%)	χ^2^ = 15.407 *p* = 0.0015
	Moderate	11	6 (22.22%)	5 (13.16%)	
	Poorly	38	11 (40.74%)	27 (71.05%)	
	Undifferentiated	8	8 (29.63%)	0 (0.00%)	
TNM staging of osteosarcoma	Stage 1	7	2 (7.41%)	5 (13.16%)	χ^2^ = 9.574 *p* = 0.0367
	Stage 2	12	6 (22.22%)	6 (15.79%)	
	Stage 3	19	4 (14.81%)	15 (39.47%)	
	Stage 4	27	15 (55.56%)	12 (31.58%)	

**Table 4 diagnostics-15-00266-t004:** Biological markers of OS patients depending on the age group.

Variable	Age Group	Mean	SD	*p*-Value	Normal Values
Alkaline Phosphatase (U/L)	<12 years	982.6	407.0	0.875	<500
	≥12 years	998.1	377.6		<420 (Female) <750 (Male)
Lactate Dehydrogenase (U/L)	<12 years	612.4	246.4	0.775	157–272
	≥12 years	629.1	221.2		
ESR (mm/h)	<12 years	67.0	25.1	0.727	18–48
	≥12 years	69.3	25.8		
Neutrophils (No/µL)	<12 years	795.9	294.8	0.035 (*)	13.000–38.000
	≥12 years	929.3	318.5		
Lymphocytes (No/µL)	<12 years	718.3	370.3	0.437	3000–9500
	≥12 years	789.2	353.5		
Platelets (No/µL)	<12 years	67.8	15.8	0.759	150.000–450.000
	≥12 years	69.1	17.5		

(*) *p* < 0.05.

**Table 5 diagnostics-15-00266-t005:** The coefficients of the model and the Wald test in the logistic regression regarding the predictive factors for the probability of death.

Multiple Regression	B	SE	Wald	Sig. *p*	Odd Ratio Exp(β)	95% CI for Exp(B)	
						Lower	Upper
Age	2.654	0.002	1.231	0.1241	**0.538**	0.587	5.819
Gender	5.537	0.271	0.847	0.7982	**0.087**	0.682	9.008
Location	7.004	0.679	2.833	0.0931	**1.007**	0.271	3.965
Tumoural Grading	3.541	0.167	18.65	0.0268 *	**2.879**	1.869	6.379
Histological Subtype	8.576	0.105	19.74	0.0187 *	**3.544**	1.886	5.871
TNM Stage	6.241	0.217	21.61	0.0042 *	**5.973**	2.671	9.822
Constant	2.844	0.194	1.69	0.0256	**1.520**	1.279	6.774

CI—confidence interval, SE—standard error; (*) Marked effects are significant at *p* < 0.05.

## Data Availability

The original contributions presented in the study are included in the article; further inquiries can be directed to the corresponding authors.

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
