# Peer review of "Clinical and Pathological Profile of Children and Adolescents with Osteosarcoma"

_diagnostics, 2025, doi:10.3390/diagnostics15030266_

Round 1
Reviewer 1 Report (Previous Reviewer 1)
Comments and Suggestions for Authors
Dear Authors
1. It is necessary to improve Fig. 1. You could add period (time) of your study and explain what does mean the abbreviation OS used in this figure. Consistency of writting should be maintained - surgical interventiona and No of surgeries. What is OS? it could be explain.
2. Table 1 - what kind of distribution did the data have and depending of these facts you should present data - mean+/-SD or median and quartiles.
3. Table 2 - in my opinion it is not necessary to include * and**, there are mistakes in the degrees of osteosarcoma differentiation and in stages. This part of the table needs corrections and improvements.
Author Response
Dear Reviewer,
Thank you very much for evaluating our manuscript. Your recommendations and comments have helped us improve our manuscript. Here we provide the requested corrections and address the comments. The changes we have made in the manuscript are highlighted in red.
- It is necessary to improve Fig. 1. You could add period (time) of your study and explain what does mean the abbreviation OS used in this figure. Consistency of writing should be maintained - surgical intervention and No of surgeries. What is OS? it could be explain.
Response: We corrected Figure 1 and added the period of time.
- Table 1 - what kind of distribution did the data have and depending of these facts you should present data - mean+/-SD or median and quartiles.
Response: Since the analyzed variable age (years) does not follow a normal distribution (the Shapiro-Wilk test returned a statistically significant result), we presented our data in Table 1 using the median values and quartiles. We added this information in the text – Results section.
- Table 2 - in my opinion it is not necessary to include * and**, there are mistakes in the degrees of osteosarcoma differentiation and in stages. This part of the table needs corrections and improvements.
Response: We corrected Table 2 (now is Table 3); we deleted the unnecessary * and**.
Reviewer 2 Report (Previous Reviewer 2)
Comments and Suggestions for Authors
Dear Authors,
I receive a second version of this particular manuscript. While I have seen some efforts in the revision, there is still much left to be desired
1) Still, regarding the aim, the authors could not "create" anything from this dataset. The authors could only describe descriptively, but there is nothing to be created here.
2) Is this a retrospective study or a cross-sectional one? A retrospective cohort means two groups have two different clinical outcomes, and age is not included in the "two different groups". This has serious implications as a cross-sectional study cannot be assessed using the survival rate. Please consult with your statistician.
3) Please include lines 97-99 in your CONSORT diagram.
4) Lines 110-112 --> Please include the citation to back up the claim of age 12 in this study
5) Please include a complete radiological report for all the X-rays, CT scans, and MRIs. Even laypeople would know that there is a mass there, but in the academic and medical world, we would like to know how extensive the tumor is.
6) The layout for Table 2 is all over the place, with the rows and columns not matching at all
7) Please include the TNM staging of stages 1-4 in the methodology section. Also, indicate which version criteria of the TNM staging the authors used
8) How was neutrophils cut-off adjusted for age? Was it adjusted individually or through the median of 12 years? Please indicate this clearly in the methodology section. If the cut off was for the whole group, please indicate the number and the citation.
9) Again, the problem with survival rate is not as easy as just inputting your data into the software. Just because you could, does not mean you should. The authors did not factor the TNM stage into the Kaplan Meier curve, the treatment, the immunohistochemistry, comorbidity, etc. The survival rate could easily be another paper if the authors have the necessary variables. Otherwise please remove it from the manuscript.
Author Response
Dear Authors, I receive a second version of this particular manuscript. While I have seen some efforts in the revision, there is still much left to be desired
Dear Reviewer,
Thank you very much for evaluating our manuscript. Your recommendations and comments have helped us improve our manuscript. Here we provide the requested corrections and address the comments. The changes we have made in the manuscript are highlighted in red.
- Still, regarding the aim, the authors could not "create" anything from this dataset. The authors could only describe descriptively, but there is nothing to be created here.
Response: We corrected the sentence in the Aim of the study section as well in the Conclusion section.
- Is this a retrospective study or a cross-sectional one? A retrospective cohort means two groups have two different clinical outcomes, and age is not included in the "two different groups". This has serious implications as a cross-sectional study cannot be assessed using the survival rate. Please consult with your statistician
Response: We conducted a retrospective study of all children and adolescents with mus-culoskeletal tumors diagnosed and treated at our Orthopedic Department. We are sorry for the mistake; we corrected it.
- Please include lines 97-99 in your CONSORT diagram.
Response: We corrected Figure 1 and added the information from lines 97-99
- Lines 110-112 --> Please include the citation to back up the claim of age 12 in this study
Response: We included references no 7-9. The two groups of pediatric patients with osteosarcoma correspond to the classification of the child (age less than 12 years) and the adolescent (age between 12-18 years). Considering the fact that age is an important variable for stratifying the risk of osteosarcoma occurrence, the analysis carried out used the age of 12 years as a reference value for the grouping of pediatric patients with osteosarcoma.
- Please include a complete radiological report for all the X-rays, CT scans, and MRIs. Even laypeople would know that there is a mass there, but in the academic and medical world, we would like to know how extensive the tumor is.
Response: We added new information - Table 2, showing the diameter of the primary tumor (cm), measured on CT scans, MRI or X-rays imaging, depending on the age, gender, and location of the tumor.
- The layout for Table 2 is all over the place, with the rows and columns not matching at all
Response: We corrected Table 2 (now is Table 3).
- Please include the TNM staging of stages 1-4 in the methodology section. Also, indicate which version criteria of the TNM staging the authors used.
Response: The American Joint Committee on Cancer (AJCC) and the Union for International Cancer Control (UICC) described the TNM staging system in 2010, which we also used in our clinic during the analyzed period. We added the information in the Methodology section and the reference number 8.
- How was neutrophils cut-off adjusted for age? Was it adjusted individually or through the median of 12 years? Please indicate this clearly in the methodology section. If the cut off was for the whole group, please indicate the number and the citation.
Response: We added the suggested information in the Methods section, as well as the citation (number 8).
- Again, the problem with survival rate is not as easy as just inputting your data into the software. Just because you could, does not mean you should. The authors did not factor the TNM stage into the Kaplan Meier curve, the treatment, the immunohistochemistry, comorbidity, etc. The survival rate could easily be another paper if the authors have the necessary variables. Otherwise please remove it from the manuscript.
Response: We deleted the Kaplan-Meier curve and a part of the manuscript referring to the survival rate.
Thank you again for evaluating our manuscript!
Reviewer 3 Report (New Reviewer)
Comments and Suggestions for Authors
The article is interesting, however, hard to read.
Major mistakes:
Methods: Add year of applied WHO diagnostic names and grading system, and year of applied AJCC staging.
Table 1. What you mean: "Rural - Mean = 13.1" - isn't it "Diameter of the
tumor (cm) in the subgroups" in the all table?
Table 2. Correct Table 2, please - f.e. you included 3x "Tumor grading Degree of osteosarcoma differentiation", and 3x "TNM staging of osteosarcoma (*) p < 0.05; (**) p < 0.01".
Table 3. Add a norm/cutoff for parameters, please.
Discussion: describe briefly: Tumor-infiltrating lymphocytes (TILs, especially cytotoxic lymphocytes), macrophage types, and neutrophils correlation in overall survivals of other mesenchymal tumours, please.
Conclusions: "..., the epidemiological and clinical profile of a child with osteosarcoma is 13 years old boy with femoral or tibia tumor, or a 12 years old girl with scapula or clavicle tumor, both coming from rural area" - according Table 2 number of tumours in scapula and clavicle is 3+3 = 6, and according to Table 1 Female = 24, why then typical girl have rare site tumour? Besides, what is overall percentage of young people living in rural area of Romania.
Minor mistakes:
You can insert an abbreviation of "Osteosarcoma" in maintext - f.e. "OS".
Fig. 1:
a) "Surgical interventions for other malignant tumors" - make * and footnote of examples - it's not clear what is a difference between interventions and surgeries;
b) "No. of surgeries for oncological and non-oncological diseases with similar OS locations" - make ** and describe "similar OS locations".
Figures 3, 4, 5, and 6 - please, make one "Figure 4. ...general.. A -..., B..., C - ..., D - ... .", you can use f.e. PowerPoint to merge pictures, and export it as tif. Similarly, in the case of Figs. 7, 8, and 9 (merge to one figure, please).
Table 3 - "Std. Dev." - previously in the article there was "SD".
Merge Fig. 10 with Table 4 - picture and at the bottom numbers of patients alive and dead. Correct "varsta" and "luni" to the "age" and "months".
Line 162: "Demographic and pathological characteristics of the studied group are presented in Table 1 and 2." - add "s" in "Tables", please. Also in the text, there are many "a" and "the" missing.
Author Response
Dear Reviewer,
Thank you very much for evaluating our manuscript. Your recommendations and comments have helped us improve our manuscript. Here we provide the requested corrections and address the comments. The changes we have made in the manuscript are highlighted in red.
- Methods: Add year of applied WHO diagnostic names and grading system, and year of applied AJCC staging.
Response: We added the suggested information in the Methods section.
- Table 1. What you mean: "Rural - Mean = 13.1" - isn't it "Diameter of the tumor (cm) in the subgroups" in the all table?
Response: Table 1 was corrected. It represents the age distribution of patients according to gender and social environment.
- Table 2. Correct Table 2, please - f.e. you included 3x "Tumor grading Degree of osteosarcoma differentiation", and 3x "TNM staging of osteosarcoma (*) p < 0.05; (**) p < 0.01".
Response: We corrected Table 2 (now is Table 3).
- Table 3. Add a norm/cutoff for parameters, please.
Response: We added te suggested information in Table 4.
5. Discussion: describe briefly: Tumor-infiltrating lymphocytes (TILs, especially cytotoxic lymphocytes), macrophage types, and neutrophils correlation in overall survivals of other mesenchymal tumours, please.
Response: We added the suggested information (and references 35, 36) in the Discussion section – lines 443-450.
6. Conclusions: "..., the epidemiological and clinical profile of a child with osteosarcoma is 13 years old boy with femoral or tibia tumor, or a 12 years old girl with scapula or clavicle tumor, both coming from rural area" - according Table 2 number of tumours in scapula and clavicle is 3+3 = 6, and according to Table 1 Female = 24, why then typical girl have rare site tumour? Besides, what is overall percentage of young people living in rural area of Romania.
Response: The number of female patients with clavicle or scapula tumor was double compared to males. We filled in the data in Table 2, and made the necessary corrections in the Conclusions chapter and in the Abstract. Regarding demographic data, there are no precise statistics for the exact number of children living in villages in Romania at any given time, but official data from censuses and official reports indicate a significant trend of rural to urban migration. According to data published by the National Institute of Statistics, approximately 45% of Romania's population lived in rural areas in 2021, and a large part of this population includes children. We included this statement in the Discussion section.
7. Minor mistakes:
You can insert an abbreviation of "Osteosarcoma" in maintext - f.e. "OS".
Response: We made this abbreviation in the second sentence of the introduction.
8. Fig.1:
a) "Surgical interventions for other malignant tumors" - make * and footnote of examples - it's not clear what is a difference between interventions and surgeries;
b) "No. of surgeries for oncological and non-oncological diseases with similar OS locations" - make ** and describe "similar OS locations".
Response: We made the suggested corrections.
9. Figures 3, 4, 5, and 6 - please, make one "Figure 4. ...general.. A -..., B..., C - ..., D - ... .", you can use f.e. PowerPoint to merge pictures, and export it as tif. Similarly, in the case of Figs. 7, 8, and 9 (merge to one figure, please).
Response: We made the suggested corrections.
10. Table 3 - "Std. Dev." - previously in the article there was "SD".
Response: We corrected.
11. Merge Fig. 10 with Table 4 - picture and at the bottom numbers of patients alive and dead. Correct "varsta" and "luni" to the "age" and "months".
Response: At the other reviewer suggestion, we deleted that part of the manuscript.
- Line 162: "Demographic and pathological characteristics of the studied group are presented in Table 1 and 2." - add "s" in "Tables", please. Also in the text, there are many "a" and "the" missing.
Response: We corrected. Thank you.
Round 2
Reviewer 1 Report (Previous Reviewer 1)
Comments and Suggestions for Authors
I don't have.
Author Response
Dear Reviewer,
Thank you again for evaluating our manuscript.
Reviewer 2 Report (Previous Reviewer 2)
Comments and Suggestions for Authors
Dear Authors,
Despite the attempts in revisions, there are still some points that are not addressed:
1) The caption of all figures are still basic, without any significant information added
2) The abstract still contains the survival rate.
3) Line 320 "post-ERCP complications", is this an ERCP procedure?
Comments on the Quality of English Language-
Author Response
Dear Reviewer,
Thank you again for evaluating our manuscript.
- The caption of all figures are still basic, without any significant information added
Response: We added information for all figures.
- The abstract still contains the survival rate.
Response: We corrected.
- Line 320 "post-ERCP complications," is this an ERCP procedure?
Response: We corrected; thank you.
Reviewer 3 Report (New Reviewer)
Comments and Suggestions for Authors The major issues in the article have been corrected well enough, however, it needs English correction. Further, lines 107-111 seem to be unnecessary in case of normal values in Table 4. The neutrophil-related changes should be comment in Discussion (line 411-416) - explain the kind of correlations (+/- and correlated parameter/s; "demonstrated that low neutrophile and platelet number is significantly associated with the prognosis of OS patients ... only a significant positive correlation of low neutrophils number with the age under 12 years could be demonstrated"), please. Methods - remove statistics tests that outcomes were not presented in the article, please.
Author Response
Dear Reviewer,
Thank you again for evaluating our manuscript.
- The major issues in the article have been corrected well enough, however, it needs English correction. Further, lines 107-111 seem to be unnecessary in case of normal values in Table 4.
Response: We corrected English. Lines 107-111 were added at another reviewer’s suggestion.
- The neutrophil-related changes should be comment in Discussion (line 411-416) - explain the kind of correlations (+/- and correlated parameter/s; "demonstrated that low neutrophile and platelet numbers are significantly associated with the prognosis of OS patients ... only a significant positive correlation of low neutrophils number with the age under 12 years could be demonstrated"), please.
Response: We discussed the neutrophil-related changes.
- Methods - remove statistics tests that outcomes were not presented in the article, please.
Response: We corrected, thank you.
Round 3
Reviewer 2 Report (Previous Reviewer 2)
Comments and Suggestions for Authors
Dear Authors,
I have no further comments.
This manuscript is a resubmission of an earlier submission. The following is a list of the peer review reports and author responses from that submission.
Round 1
Reviewer 1 Report
Comments and Suggestions for Authors
Dear Authors
1. why do you exclude patients with previous oncological treatment for another reason?
2. Fig. 1 - I don't understand. How many patients and in which age (limits) do you analyze?
3. Fig. 2 - is not necessary - only this with mean+/-SD
4. Table 1 - why didn't you compare gender and environment between patients < 12 years and >12 years of age?
5. Fig. 5 is not clear.
6. All sections of your article need corrections.
7. I'm not sure that based on the analyse of 65 patients we are able to establish the epidemiological and clinical profile of a child with OSA
Reviewer 2 Report
Comments and Suggestions for Authors
Dear Authors, I have read a manuscript about the clinical and pathological profiles of children and adolescents with osteosarcoma. While this is a commendable effort, several aspects need to be addressed before this paper can be considered for publication.
Overall English Language
The English language is not bad per se, but several terms are too informal for a proper paper, such as "Thanks to the remarkable" or "Anyway, there is....(line 77)" and some typos, such as "sage IV TNM" on line 42. Therefore, I strongly suggest that the authors rephrase some of the inappropriate words and look for grammatical errors.
Introduction
After reading the whole paper, I realized that this study could not create a clinical and pathological profile as it is basically just a descriptive study. In order to develop a clinical and pathological profile, the authors would need an extensive database with more detailed data, such as the molecular histopathology analysis. Hence, please revise the aim and change all the conclusions, including those in the abstract section.
Methods
1) When the authors say diagnosed and treated for osteosarcoma, how was the diagnosis confirmed? Were all patients confirmed by biopsy or through surgical-pathological analysis? Please specify.
2) Why were patients who were initially operated elsewhere excluded? I see little justification for doing so if all the data for the analysis are complete.
3) What do the authors mean by "other than OS musculoskeletal tumor?"
4) The authors mentioned that all patients are investigated by X-ray, CT, bone scan, and MRI. Please include some figures as well as the results in the tables.
5) Please specify when the blood study was done. Was it initially for diagnosis, pre-operatively, or post-operatively?
6) How many cases are adult cases? Please include this in Figure 1 so readers can estimate how many pediatric patients there are in your hospital.
7) The authors divided the cohort into <12 years and >12 years without any prior justification. The authors mentioned the reasoning later in the results section, but this reasoning should be provided in the methods section.
8) How many pathologist anatomists grade the tumors?
9) What was the TNM staging used in this study? Also, it is unusual to use grades 1-4. Usually, TNM staging is stages 1-4.
10) What is the methodology and sampling method of this study? Was this a cross-sectional one? Cohort? Was this a consecutive sampling study?
Results
1) There is no need to present both median and mean if the authors already provided.
2) Figures 2, 5, 6, and 7 are redundant. Please remove them as they can easily be described through one or two sentences
3) Please provide more descriptions in Figures 3 and 4 instead of just "humeral or femoral OS". Please describe the patient's age, presentation, size, initial imaging, and outcome.
4) The authors mentioned the size of the tumors, but there is no data on such in Table 1. Please include that in Table 1, as well as how was the tumor measured. Through imaging or surgical measurement?
5) The number of genders and social environments are left empty in Table 1. Are the authors sure that all have agreed and that they completely missed the empty rows in Table 1?
6) As for Table 2, neutrophils cut-off are sensitive to age. Has this value been adjusted for age?
7) The authors also mention the survival rate. In order to calculate the survival rate correctly, the authors have to adjust the initial presentation, the stage, and the treatment. Furthermore, this is not part of the aim of the study. Usually, a survival study would become a separate analysis and a separate paper. I suggest removing this section.
8) What is an "ENTER" model? This analysis should be specified in the methods section.
Comments on the Quality of English LanguageAlready discussed in the comments and suggestions for the authors
